# Discovery of AI-2 Quorum Sensing Inhibitors Targeting the LsrK/HPr Protein–Protein Interaction Site by Molecular Dynamics Simulation, Virtual Screening, and Bioassay Evaluation

**DOI:** 10.3390/ph16050737

**Published:** 2023-05-12

**Authors:** Yijie Xu, Chunlan Zeng, Huiqi Wen, Qianqian Shi, Xu Zhao, Qingbin Meng, Xingzhou Li, Junhai Xiao

**Affiliations:** 1National Engineering Research Center for Strategic Drugs, Beijing Institute of Pharmacology and Toxicology, Beijing 100850, China; beilou2353@163.com (Y.X.);; 2State Key Laboratory of Toxicology and Medical Countermeasures, Beijing Institute of Pharmacology and Toxicology, Beijing 100850, China; xyz18359103933@163.com (C.Z.);; 3State Key Laboratory of Pathogen and Biosecurity, Institute of Microbiology and Epidemiology, Academy of Military Medical Sciences, Beijing 100071, China; 4Department of Hepatology, Fifth Medical Center of Chinese PLA General Hospital, Beijing 100039, China

**Keywords:** AI-2, quorum sensing, antibacterial agents, LsrK, virtual screening, molecular dynamics, quorum sensing inhibitors, HPr

## Abstract

Quorum sensing (QS) is a cell-to-cell communication mechanism that regulates bacterial pathogenicity, biofilm formation, and antibiotic sensitivity. Among the identified quorum sensing, AI-2 QS exists in both Gram-negative and Gram-positive bacteria and is responsible for interspecies communication. Recent studies have highlighted the connection between the phosphotransferase system (PTS) and AI-2 QS, with this link being associated with protein-protein interaction (PPI) between HPr and LsrK. Here, we first discovered several AI-2 QSIs targeting the LsrK/HPr PPI site through molecular dynamics (MD) simulation, virtual screening, and bioassay evaluation. Of the 62 compounds purchased, eight compounds demonstrated significant inhibition in LsrK-based assays and AI-2 QS interference assays. Surface plasmon resonance (SPR) analysis confirmed that the hit compound 4171-0375 specifically bound to the LsrK-N protein (HPr binding domain, KD = 2.51 × 10^−5^ M), and therefore the LsrK/HPr PPI site. The structure-activity relationships (SARs) emphasized the importance of hydrophobic interactions with the hydrophobic pocket and hydrogen bonds or salt bridges with key residues of LsrK for LsrK/HPr PPI inhibitors. These new AI-2 QSIs, especially 4171-0375, exhibited novel structures, significant LsrK inhibition, and were suitable for structural modification to search for more effective AI-2 QSIs.

## 1. Introduction

Bacterial infection, especially involving multidrug-resistant bacteria, remains a major risk factor for human disease and death [1]. The increasingly weak therapeutic effect of antibiotics against multidrug-resistant bacteria and the emergence of more widely resistant multidrug-resistant bacteria pose a considerable threat to human health [2]. The use of traditional antibiotics to inhibit the growth or survival of bacteria also leads to the emergence of drug-resistant bacteria, whether the drug-resistant genes are mutated under antibiotic pressure or spontaneously arise. However, the infection does not result just from the existence of pathogens but also depends on the interaction between the pathogen and the host. To date, several new therapies have been developed to achieve therapeutic effects while avoiding bacterial resistance [3]. Among them, quorum sensing inhibitors (QSIs), a new type of inhibitors, are considered to have the potential of reducing the virulence, motility, and biofilm formation of bacteria without affecting bacterial growth and survival, and thus transform pathogenic bacteria into non-pathogenic bacteria [4,5].

Quorum sensing (QS) is a process by which bacteria communicate and respond to changes in their population density through the secretion and detection of small-molecule signaling molecules called autoinducers (AIs) [6,7]. This mechanism allows bacterial cells to coordinate their behavior and regulate gene expression based on the density of nearby cells [8,9]. The diffusion mechanism of the signaling molecules is crucial for this process, as it enables the bacteria to sense changes in the concentration of AIs in the environment and respond accordingly. When the concentration of AIs reaches a certain threshold, it triggers the expression of QS-related genes, leading to the modulation of various physiological processes, such as bioluminescence [10], virulence production [4], biofilm formation [11], susceptibility to antibiotics [12], and motility [13]. At present, widely studied AIs include acylated homoserine lactones (AHLs) [14,15,16], autoinducer-2 (AI-2), autoinducer peptides (AIPs) [17,18,19], and other AIs [20,21]. Among them, AI-2 QS system exists in both Gram-negative and Gram-positive bacteria and is detected in more than 55 species, including *Escherichia coli*, *Salmonella typhimurium*, *Vibrio harveyi*, and *Klebsiella pneumoniae* [22].

The AI-2 signal is not a single signaling molecule but a group of (4*S*)-4,5-dihydroxy-2,3-pentanedione ((*S*)-DPD) derivatives, produced by the MetK, Pfs, and LuxS enzymes in turn (Figure 1). Specifically, *S*-adenosylhomocysteine (SAH) is generated from *S*-adenosylmethionine (SAM) through demethylation mediated by MetK, then Pfs removes adenine from SAH to form *S*-ribosylhomocysteine (SRH). Cleavage of homocysteine in SRH by LuxS produces *S*-4,5-dihydroxy-2,3-pentanedione ((*S*)-DPD) [23]. After being transferred to the outside of the cell [24], (*S*)-DPD can form *S*-DHMF and *R*-DHMF reversibly by self-cyclization. Subsequently, *S*-THMF and *R*-THMF can be reversibly formed from *S*-DHMF and *R*-DHMF by hydration [25]. When the concentration of extracellular DPD molecules reaches a certain threshold, it can promote DPD molecules to re-enter the bacteria by binding with LuxP [26], LsrB [27], or RbsB [28]. Here, we focus on the AI-2 QS system with the LsrACDB transport system. After DPD is transported into the cell through the membrane channel formed by LsrACDB [29], it binds to LsrK in its linear form and is phosphorylated into P-DPD [30]. If P-DPD binds to LsrR [30], LsrR would no longer suppress the transcription of the *lsr* operon and processing of DPD was accelerated, thus promoting the positive regulation of AI-2 QS [31]. LrsK is therefore an important functional protein that phosphorylates DPD molecules in AI-2 QS. Knockout of the *lsrK* gene greatly reduced internalization of DPD, which led to the suppression of *lsr* operon [32]. Previous studies certified that AI-2 QS is inhibited in a glucose-rich environment, and a recent study [33] reported that this relationship is related to the interactions between LsrK and HPr (one of the important members of PTS). When glucose is deficient, the His 15 residue of the HPr mainly exists in a phosphorylated state and the p-HPr weakly inhibits the phosphorylation function of LsrK. However, when glucose is sufficient, the His 15 residue of the HPr remains mainly unphosphorylated and HPr significantly inhibits the phosphorylation function of LsrK. Therefore, it is clear that if small molecule inhibitors could be found to mimic the binding of HPr to LsrK, it would be possible to inhibit the phosphorylation of LsrK and AI-2 QS. As far as we know, the currently reported AI-2 QSIs targeting LsrK are mainly ATP competitive inhibitors or DPD Analogues [34,35,36,37], while there are no reports of AI-2 QSIs targeting the LsrK/HPr PPI site.

In this study, we confirmed a suitable binding site for small molecules at the LsrK/HPr PPI site by long-term MD simulation of the LsrK/HPr crystal structure. LsrK inhibitors targeting the LsrK/HPr PPI site were searched by in silico virtual screening of large-scale commercial compound libraries combined with biological assays. SPR analysis confirmed that the hit compound specifically bound to the LsrK-N protein (HPr binding domain, residues 1-260), and therefore the LsrK/HPr PPI site. Several AI-2 QSIs with novel structures targeting the LsrK/HPr PPI site were discovered and the LsrK/HPr PPI site was confirmed as a new target of AI-2 QSIs.

## 2. Results and Discussion

### 2.1. Analysis of the LsrK-HPr PPI Site

The PPI site was difficult to target by small molecules because of its large interaction area and dispersion of key amino acid residues [38]. So, we first analyzed whether the LsrK/HPr PPI site was suitable for the binding of small-molecule inhibitors. The LsrK/HPr PPI site in the crystal structure (PDB: 5YA1, Figure 2A) was visualized and evaluated by “Binding Site Detection” (Schrödinger 2020-3) and this confirmed the presence of a potential small-molecular binding site with a Sitescore = 0.88 and Dscore = 1.01. Then, the interactions between LsrK and HPr were analyzed by “Protein Interaction Analysis” (Schrödinger 2020-3). The results demonstrated that there were multiple types of interactions between the LsrK and HPr, including hydrogen bonds, salt bridges, and hydrophobic interactions (Appendix A). The residues of LsrK (such as Glu 122, Asp 160, and Arg 163) formed hydrogen bonds or salt bridges with residues of Hpr (such as Hip 15, Lys 40, and Lys 49) (Figure 2B). A hydrophobic pocket was observed at the LsrK/HPr PPI site, and the key polar residues were distributed on both sides of the hydrophobic pockets. Strong hydrophobic interactions were maintained by Leu 47 and Phe 48 of the HPr and the hydrophobic pocket of LsrK was formed by Leu 123, Leu 126, Leu 152, Tyr 162, Met 210, and Ala 211 residues (Figure 2). Taken together, these findings indicated that the LsrK/HPr PPI site was suitable for small-molecule inhibitor binding.

To further understand the dynamic interactions of the LsrK/HPr complex and determine the key residues, a 1000 ns MD simulation of the LsrK/HPr complex was carried out using GROMACS 2021.3. The root mean square deviation (RMSD) was used to evaluate the fluctuation of the backbone of the LsrK/HPr complex. RMSD values (Figure 3A) indicated that the backbone of the LsrK/HPr complex was in equilibrium during the subsequent 300–1000 ns, except for a small fluctuation during 0–300 ns. The average structures of the LsrK/HPr complex were derived in the unit of 10 ns for visual inspection. The observation results demonstrated that fluctuation of the LsrK/HPr backbone was caused by clockwise rotation of the C-terminal domain of LsrK, rather than the HPr binding domain (N-terminal domain). 

20,000 frames from 800–1000 ns were selected to calculate the binding free energy and free energy decomposition of the LsrK/HPr complex by the molecular mechanics/Poisson–Boltzmann surface area (MM/PBSA) method. The calculated binding free energy was −64.49 ± 8.89 kcal/mol (Figure 3B). Free energy decomposition was used to evaluate the per-residue contribution to the binding free energy. The results indicated that the interactions between the hydrophobic pocket of LsrK and the Leu 47 and Phe 48 residues of HPr contributed nearly −16.5 kcal/mol to the calculated binding free energy, and they were largely dominated by Van der Waals interactions (Figure 3C,D). Consistent with our previous analysis, the hydrophobic interactions between LsrK and HPr were critical to maintain their binding. Furthermore, the hydrogen bonds and electrostatic interactions between LsrK and HPr were analyzed. The interaction region formed by the Glu 122, Glu125, and His 156 residues of LsrK and the His 15, Thr 16, and Arg 17 residues of HPr contributed to a binding free energy of nearly −13.0 kcal/mol. Another interaction region formed by the Ser 159, Asp 160, and Arg 163 residues of LsrK and the Lys 40, Gln 51, and Thr 52 residues of HPr contributed to a binding free energy of nearly −11.5 kcal/mol (Figure 3C). Energy contributed by each residue was decomposed into polar solvation, electrostatic, and Van der Waals components (Figure 3D). The electrostatic energy of the Glu 122, Glu 125, Asp 160, Ala 165, and Met 210 residues of LsrK and the His 15, Arg 17, Lys 27, Lys 40, Ser 46, Lys 49, and Thr 52 residues of HPr contributed greatly to the binding of LsrK and HPr. Residue Asp 209 was not conducive to the interaction between LsrK and HPr because of its large polar solvation energy. The energy decomposition of residue Met 210 demonstrated that it contributed to the binding of LsrK and HPr through van der Waals forces and electrostatic interaction (Figure 3D), of which the electrostatic interaction was rooted in a stable hydrogen bond formed by the Phe 48 and Met 210 residues.

In summary, MD simulation analysis also clarified that hydrophobic amino acid residues (Leu 123, Leu 126, Leu 152, Tyr 162, Ala 165, Met 210, and Ala 211) and polar amino acid residues (Glu 122, Glu 125, His 156, Asp 160, and Arg 163) played a key role in the binding of LsrK and HPr, and were also conducive to the binding of small molecule inhibitors to LsrK.

### 2.2. Virtual Screening Workflow (VSW)

Based on the crystal structure of the LsrK/HPr complex (PDB: 5YA1), virtual screening was conducted by the VSW (Schrödinger 2021.3) to search for potential LsrK/HPr PPI inhibitors from the commercial compound library ChemDiv (1.6 million molecules). In particular, according to the results of MD simulation, a constraint of forming at least one hydrogen bond with residues Glu 122, Glu 125, Asp 160, Arg 163, Ala 165, or Met 210 was introduced to improve the hit rates of virtual screening. A total of 1942 compounds passed the VSW of three precision docking cascades provided by the Schrödinger suite: high-throughput virtual screening (HTVS) precision, standard precision (SP), and extra precision (XP) screening. Among them, the top 10% of compounds based on the docking scores were selected for higher-precision virtual screening. The top 300 compounds with an XP Gscore below −7.0 kcal/mol were classified into 100 clusters according to their volume overlap using the ligand clustering module (Schrödinger 2020-3). Considering the binding diversity with key residues and ligand efficiency, 62 commercially available compounds were selected for bioassay evaluation (Appendix A).

### 2.3. LsrK Inhibition Assay

Preliminary screening was conducted by determining the LsrK inhibition of these 62 selected compounds at 200 μM (Appendix A). From these 62 compounds, 8 hits demonstrated LsrK inhibition >50%, among which compound 4171-0375 had the highest inhibition of 98 ± 7% (Table 1 and Figure 4). The IC_50_ values of these eight compounds were determined by a dose-response assay. Compound 4171-0375 demonstrated the highest LsrK inhibition with an IC_50_ = 26.13 ± 2.94 μM. To confirm whether compounds were non-ATP competitive inhibitors, three of these eight compounds (4171-0375, 4929-0003, and K659-0421) with IC_50_ values < 100 μM were tested to determine their IC_50_ values at different ATP concentrations [35]. The dose-response curves of the tested compounds indicated no significant difference at different ATP concentrations (Figure 5), which preliminarily demonstrated the non-ATP competitive characteristics of the compounds.

### 2.4. Cell-Based AI-2-Mediated QS Interference Assay

Eight compounds that demonstrated LsrK inhibition greater than 50% were selected for a cell-based AI-2 QS interference assay. All of these compounds revealed low-micromolar AI-2 QS inhibition, although some compounds demonstrated relatively low activity in the LsrK inhibition assay (Table 1). At the same time, the effects of each compound on WHQ01 (*S. typhimurium* ATCC 202165 ΔTolC pWHQ01) and WHQ02 (*E. coli* BL21 ΔTolC pWHQ01) were similar (Appendix A). Considering that the key residues at the LsrK/HPr PPI site are conserved among different bacteria [33], the results were indeed not surprising. Among the compounds, K659-0421 presented the highest AI-2 QS inhibition with an IC_50_ of 7.97 ± 0.38 μM in WHQ01 (Appendix A). Compound 4171-0375, which indicated the highest inhibition in an LsrK inhibition assay, performed poorly in the AI-2 QS interference assay, with IC_50_ values of 42.93 ± 1.63 μM against WHQ01 (*S. typhimurium* ATCC 202165 ΔTolC pWHQ01) and 29.12 ± 1.23 μM against WHQ02 (*E. coli* BL21 ΔTolC pWHQ01). Considering that the guanidine group in compound 4171-0375 was highly protonated under physiological conditions [39], the resulting non-specific binding and low membrane permeability may be reasons for the poor inhibition of AI-2 QS. Although three compounds inhibited bacterial growth at high concentrations, they still exhibited significant AI-2 QS inhibition at concentrations that did not induce obvious growth inhibition (AI-2 QS inhibition > 80%, bacterial growth inhibition < 40%). In addition, all compounds screened by the LsrK inhibition assay displayed obvious AI-2 QS inhibition, especially those without bacterial growth inhibition. Taken together, these findings indicated that the compounds identified by screening inhibited AI-2 QS.

### 2.5. SPR Assay

Bioassay results confirmed that the eight compounds identified by screening had inhibitory activity against LsrK and AI-2 QS. Further analysis identified 4171-0375, 4929-0003, and K659-0421 as non-ATP competitive inhibitors. However, there was no evidence that compounds directly bound to the LsrK/HPr PPI site. We attempted to obtain the crystal structure of LsrK/4171-0375 but were unsuccessful. Therefore, we independently expressed the LsrK-N protein (HPr-binding domain of LsrK, residues 1-260) and the specific binding between 4171-0375 (with the highest inhibition rate in the LsrK inhibition assay) and LsrK-N protein was tested by an SPR assay. First, the affinity of HPr to LsrK and LsrK-N was tested to verify the activity of LsrK-N. The KD values of LsrK and LsrK-N were 7.05 × 10^−8^ M and 9.77 × 10^−8^ M, respectively (Figure 6). Further, the affinity of 4171-0375 to LsrK and LsrK-N was tested, and the KD values were 9.45 × 10^−5^ M and 2.51 × 10^−5^ M, respectively (Figure 6). The results demonstrated that 4171-0375 bound to LsrK and LsrK-N with similar affinity, which indicated that 4171-0375 can specifically bind to LsrK-N. Considering that LsrK-N had no other suitable binding sites for small molecules, we concluded that 4171-0375 was specifically bound to the LsrK/HPr PPI site.

In summary, 8 of the 62 compounds selected by virtual screening displayed LsrK inhibition greater than 50% and three compounds (4171-0375, 4929-0003, and K659-0421), with an IC_50_ below 100 μM, were proven to be non-ATP competitive inhibitors. These eight compounds were tested in a cell-based AI-2 QS interference assay and all demonstrated obvious AI-2 QS inhibition against both WHQ01 and WHQ02. The SPR assay revealed that 4171-0375 bound to LsrK and LsrK-N, with approximate KD values determined, which indirectly indicated that 4171-0375 specifically bound to the LsrK/HPr PPI site. Experimental results confirmed that the LsrK/HPr PPI site was the target of AI-2 QSIs, and AI-2 QSIs targeting the LsrK/HPr PPI site were discovered for the first time.

### 2.6. Exploration of Structure-Activity Relationships (SARs)

We investigated the SARs of the hit compounds. Several common features were discovered in the structures of the eight hit compounds with obvious LsrK inhibition and AI-2 QS inhibition. First, they all contained benzene ring substituents, linkers, and polar substituents consisting of basic or acidic groups (Figure 4). Combined with the results of ligand docking, which were subjected to XP precision in the virtual screening, we found that the hydrophobic groups of all hit compounds were inserted into the hydrophobic pockets at the LsrK/HPr PPI site (Figure 7A). Meanwhile, the plane of the linker and the hydrophobic group was close to vertical and the ether bond, aliphatic chain, or sulfonamide bond of the linker connected to the hydrophobic group played a key role. The polar groups of compounds formed hydrogen bonds or salt bridges with different residues (Figure 7A). Based on the different interactions of polar groups with LsrK, the eight positive hit compounds could be divided into three types (Figure 8). Compounds of Type I (4171-0375, 4929-0003, and K659-0421) formed hydrogen bonds and salt bridges with Glu 125 or Glu 122 residues, compounds of Type II (3229-1889, 8012-5390, and D715-0257) formed hydrogen bonds and salt bridges with residue Arg 163, and compounds of Type III (8020-4294 and G268-0878) formed hydrogen bonds with residue Ala 165. Considering that all compounds of Type I indicated higher inhibition in the LsrK inhibition assay, these three compounds were selected to generate a simple pharmacophore model using the PHASE module (Schrödinger 2020-3) to demonstrate the structural features. The best model (PhaseHypoScore = 1.24) was APRR (Figure 7B), which comprised one hydrogen acceptor (A), one positive ionic site (D), and two aromatic rings (R). This model further validated the importance of hydrophobic groups and basic polar groups, and provided a reference for structural modification based on these three compounds. At the same time, we noted that there was a poor correlation between the XP Gscore and bioassay results. We identified two defects in ligand docking based on the crystal structure of the LsrK/HPr complex, one was that the side chains of the key polar residues were flexible, meaning that the docking score and docking poses may not be accurate. The second was that the conformation used for docking is LsrK combined with HPr, which is likely to be different from the conformation when small molecular inhibitors are combined. Taking all of this into consideration, our results provide a clear SAR, and indicate that the hydrophobic interaction with hydrophobic pockets and the existence of polar substituents are crucial to the inhibitory activity of LsrK/HPr PPI inhibitors.

## 3. Materials and Methods

All chemicals were purchased from Sigma-Aldrich (St. Louis, MO, USA) if not otherwise stated. DPD was purchased from SHANGHAI ZZBIO Co., Ltd. (Shanghai, China). The Kinase-Glo Max Luminescent kinase assay kit was purchased from Promega Corp. (Madison, WI, USA). The NTA sensor chip was purchased from GE Healthcare (Chicago, IL, USA). All compounds selected by virtual screening were purchased from Topscience Biotechnology Co., Ltd. (Shanghai, China). The QS reporter strains WHQ01 (*S. typhimurium* ATCC 202165 ΔTolC pWHQ01) and WHQ02 (*E. coli* BL21 ΔTolC pWHQ01) were donated by Huiqi Wen (Institute of Microbiology and Epidemiology, Academy of Military Sciences, Beijing, China). The plasmid pWHQ01 was constructed by cloning of *lsr* promoters and the *luxCDABE* luminescent gene. *Bam*HI, *Xho*I, *Eco*RI, and *Not*I restriction endonucleases were purchased from New England Biolabs (Ipswich, MA, USA). pET-28a and pGEX-4T-1 were purchased from Novagen (Madison, WI, USA). *E. coli* BL21 (DE3) was purchased from TransGen Biotech Co., Ltd. (Beijing, China). The Ni-NTA column and GST Gravity column were purchased from Sangon Biotech (Shanghai, China). Sodium dodecyl sulfate-polyacrylamide gel electrophoresis (SDS-PAGE) gels and the Bradford Protein Assay Kit were purchased from Thermo Fisher Scientific (Waltham, MA, USA). Ultrafiltration centrifuge tubes were purchased from Millipore (Bedford, MA, USA). Dialysis bags were purchased from Beijing Solarbio Science and Technology Co., Ltd. (Beijing, China).

### 3.1. MD Simulation, Virtual Screening and Pharmacophore Hypothesis Generation 

#### 3.1.1. Protein Preparation and Binding Site Detection

The crystal structure of the LsrK/HPr complex (PDB: 5YA1) was prepared using the “Protein Preparation Wizard” (Schrödinger 2020-3). Specifically, hydrogen atoms were added, the missing loops and side chains were filled in, the hydrogen bonding network was optimized, and restrained energy minimization (only the hydrogens) were minimized using the OPLS3 force field. All water molecules, crystallization solvents, and ATP were deleted. Protein_chain_A (LsrK) was extracted from the prepared structure of the LsrK/HPr complex. It was then used to visualize and evaluate small-molecule binding sites of LsrK by “SiteMap” (Schrödinger 2020-3) with settings specifying the reporting of up to seven sites in the standard grid. The receptor grid for virtual screening was generated using “Receptor Grid Generation” in the “Glide” module (Schrödinger 2020-3). Leu 152, Tyr 162, Glu 122, Glu 125, and Arg 163 residues were selected to define the active site for virtual screening. The box was set to 16′, 16′, 16 Å. Then, protein_chain_A (LsrK) and protein_chain_C (HPr) were extracted together from the prepared structure of the LsrK/HPr complex for MD simulation.

#### 3.1.2. Molecular Dynamics (MD) Simulation

MD simulations were performed using the software GROMACS 2021.3 [40]. The prepared LsrK/HPr complex was applied in the AMBER 99SB-ILDN force field [41]. The octahedron box dimensions for periodic boundary conditions, while maintaining a minimum distance from any atom to the boundary of the box of 1 nm, were calculated to be 8.8 × 8.0 × 6.4 nm. The TIP4P water model was used to conduct the MD simulations in explicit solvation. Sodium ions (Na^+^) were added to the LsrK/HPr system for neutralization. The steepest descent algorithm was used for energy minimization, and the maximum force was set not to exceed 1000 kJ/mol/nm. A Berendsen thermostat [42] and Parrinello-Rahman barostat [43] were used for temperature and pressure coupling, respectively. The system was equilibrated at a temperature of 300 K and 1 bar pressure by two consecutive 1000 ps simulations with canonical NVT ensembles and isobaric NPT ensembles, respectively. MD simulations were run for 1000 ns at stable temperature and pressure with a time step of 2 fs and a long-range interaction cutoff of 1 nm. After completion of the simulation, the root-mean-square deviation (RMSD) was calculated using the utilities of the GROMACS packages. We conducted three identical MD simulations using GROMACS 2021.3 with consistent simulation conditions and parameters. The results from all three replicates were similar, demonstrating the robustness and stability of our simulations.

#### 3.1.3. Estimation and Decomposition of the Binding Free Energy by gmx_MMPBSA

A molecular mechanics/Poisson–Boltzmann surface area (MM/PBSA) in gmx_MMPBSA tools [44] was applied to determine the thermodynamic stability of the LsrK/HPr complex and inspect the contribution of each residue at the LsrK/HPr PPI site. A total of 100,000 frames of the LsrK/ATP complex after equilibrium were produced and 20,000 frames over 800–1000 ns were selected to calculate the binding free energy and the per-residue free energy decomposition. The default parameters were applied for all calculations.

#### 3.1.4. Virtual Screening

The receptor grid was generated as described in Section 3.1.1. Nearly 1.6 million molecules obtained from ChemDiv, including discovery chemistry (DC), new chemistry (NC), and innovative chemistry (IC) libraries were processed by LigPrep (Schrödinger 2020-3) with default parameters to develop a virtual compound library composed of three-dimensional (3D) conformations.

According to the MD simulation of the LsrK/HPr complex, a restriction was introduced in the virtual screening that at least one hydrogen bond must be formed with a key amino acid residue (Glu 122, Glu 125, His 156, Asp 160, Tyr 162, Arg 163, or Ala 165). The newly generated compound 3D structure database was subjected to a “Virtual Screening Workflow (VSW)” of the “Glide” module (Schrödinger 2020-3) processing to obtain the initial hits. The VSW is a hierarchical multi-precision docking protocol involving three levels of increasing docking precision: high-throughput virtual screening (HTVS), standard precision (SP), and extra precision (XP). The compounds were limited with an output score of the top 10% in each stage of HTVS, SP, and XP to the next round of docking. Based on the XP Gscore, the top 300 molecules were selected and classified into 100 clusters by volume overlap. Considering the binding diversity of LsrK and ligand efficiency, potential LsrK/HPr PPI inhibitors were selected from 100 clusters by visual inspection, and finally, 62 compounds were selected and purchased for bioassay evaluation.

#### 3.1.5. Pharmacophore Hypothesis Generation

The pharmacophore hypothesis was developed by the PHASE module through three positive ligands (4171-0375, 4929-0003, and K659-0421). Multiple ligands were used to create a pharmacophore model, and the conformations of these three compounds, which were generated in virtual screening with XP precision, were defined as actives. The pharmacophore method was then used to identify the best alignment and common features. All other parameters were set to default.

### 3.2. Bioassay Methods

#### 3.2.1. Overexpression and Purification of LsrK, LsrK-N, and HPr 

To express LsrK or LsrK-N with a hexa-histidine tag at the C-terminus, the LsrK or LsrK-N encoding gene of *S. typhimurium* was amplified using primers carrying restriction endonuclease sites for *Bam*HI and *Xho*I, respectively. The amplified product was cloned into the expression vector pET-28a with a C-terminal hexa-histidine tag and transfected into *E. coli* BL21(DE3) cells. The transformed cells were selected on LB agar plates containing 50 µg/mL kanamycin. The recombinant isolates were grown at 37 °C to the mid-log exponential phase. Protein synthesis was induced by 0.5 mM IPTG. After overnight growth, the cells were pelleted at 13,000× *g* for 10 min at 4 °C, resuspended in lysis buffer (50 mM NaH_2_PO_4_, 300 mM NaCl, and pH 8.0), and sonicated on ice for 10 cycles with a 30-s pulse and 30-s pause. The bacterial lysate was then centrifuged at 4 °C, 13,000× *g* for 10 min, and the supernatant passed through a 0.22 µm filter. 

The filtrate was loaded onto a Ni-NTA column, and proteins were eluted with 5 volumes of imidazole-containing buffer (50 mM NaH_2_PO_4_, 300 mM NaCl, 250 mM imidazole, and pH 8.0) via a step gradient to recover the purified LsrK. The latter was then dialyzed overnight in a bag of 55 kDa molecular-mass-cutoff membrane for LsrK and a bag of 14 kDa molecular-mass-cutoff membrane for LsrK-N, against lysis buffer at 4 °C. The molecular weight of LsrK or LsrK-N was determined by SDS-PAGE followed by staining with Coomassie blue. LsrK or LsrK-N was then quantified using the Bradford Protein Assay Kit. 

To express HPr with a GST tag at the N-terminus, the HPr encoding gene from *S. typhimurium* was amplified using primers carrying restriction endonuclease sites for *Eco*RI and *Not*I, respectively. The amplified product was cloned into the expression vector pGEX-4T-1 with an N-terminal GST tag and transfected into *E. coli* BL21(DE3) cells. The transformed cells were selected on LB agar plates containing 100 µg/mL Ampicillin. The recombinant isolates were grown at 37 °C to the mid-log exponential phase. Protein synthesis was induced by 0.5 mM IPTG. After overnight growth, the cells were pelleted at 13,000× *g* for 10 min at 4 °C, resuspended in buffer A (500 mM NaCl, 25 mM Tris-HCl, 1 mM 1,4-Dithiothreitol (DTT), 0.1% (*v*/*v*) Triton X-100, and pH 8.0), and sonicated on ice for 10 cycles with a 30-s pulse and 30-s pause. The bacterial lysate was then centrifuged at 4 °C, 13,000× *g* for 10 min, and the supernatant passed through a 0.22 µm filter. 

The filtrate was loaded onto a GST Gravity column, and proteins were eluted with 5 volumes of buffer B (500 mM NaCl, 25 mM Tris-HCl, 10 mM Glutathione (GSH), 1 mM DTT, and pH 8.0) to recover the purified HPr. The latter was then loaded onto a Ultrafiltration centrifuge tube of 10 kDa, then centrifuged at 4 °C, 14,000× *g* for 10 min, and resuspend in the elution buffer (150 mM NaCl, 25 mM Tris-HCl, 1 mM TCEP, 1 mM DTT, and 5% (*v*/*v*) Glycerine) to concentrate the HPr. The molecular weight of HPr was determined by SDS-PAGE followed by staining with Coomassie blue. HPr was then quantified using the Bradford Protein Assay Kit. All proteins were concentrated and then stored at −80 °C.

#### 3.2.2. LsrK Inhibition Assay

The LsrK inhibition assay was performed as described previously [35]. The 62 selected compounds were dissolved in DMSO at a concentration of 2 mM. The assay was carried out with final concentrations of 300 nM LsrK, 100 μM ATP, and 300 μM DPD in a reaction buffer containing 25 mM TEA, pH 7.4, 800 μM MgCl_2_, and 0.1 mg/mL BSA. Specifically, 10 μL DPD, 10 μL LsrK, and 5 μL of the compound were added to a 96-well plate, and a 25 μL ATP or 25 μL reaction buffer was added after 30 min of incubation. The plate was incubated for 10 min at 37 °C, then 50 μL of kit reagent was added and the plate was incubated at 37 °C for 15 min. The luminescence signal was measured by the Enspire 2300 microplate reader (PerkinElmer). To confirm the activity of hits selected by primary screening, dose-response experiments were performed (200–6.25 µM). The inhibition % for each tested compound was determined as described previously [35]. The IC_50_ values were calculated using the four logistic parameters. 

#### 3.2.3. SPR Assay

The His-tagged LsrK or LsrK-N protein was immobilized in the NTA sensor chip (GE Healthcare) in the Biacore 8K through a running buffer consisting of 1× PBS-T. Serial dilutions of HPr were injected with the concentration ranging from 600 to 18.7 nM. When testing the affinity of 4171-0375 with LsrK or LsrK-N protein, the running buffer consisted of 1× PBS-T and 5% *v*/*v* DMSO. Serial dilutions of compound 4171-0375 were injected, ranging in concentrations from 100 to 1.56 µM, an intermediate concentration of 12.5 µM was set as a repeat, and 1× PBS-T with 5% *v*/*v* DMSO was set as a control group. The association time was set to 120 s, the dissociation time was set to 130 s, and the flow rate was set to 30 µL/min. The resulting data were fit to the affinity binding model using Biacore Evaluation Software (GE Healthcare).

#### 3.2.4. Cell-Based AI-2-Mediated QS Interference Assay

WHQ01 or WHQ02 was cultured overnight and diluted 1:100 in fresh LB supplemented with 100 µg/mL ampicillin and grown at 30 °C until the logarithmic phase (OD_600_ = 0.4–0.6). The bacterial culture was centrifuged at 4000× *g* for 5 min and the pellet was resuspended with fresh LB to prepare a suspension of 2 McFarland standard, which was added to a 96-well plate. In the experimental group, 62 compounds with final concentrations ranging from 100 to 1.53 μM were added to obtain a total volume of 100 µL. A negative control comprising 100 µL of LB with DMSO of the same volume and a positive control comprising 100 µL of LB supplemented with 2% glucose and DMSO of the same volume were also added. Then, bacteria were grown at 37 °C for 3 h. The fluorescent intensity of WHQ01 or WHQ02 was directly regulated by the Lsr promoters to reflect the strength of AI-2 QS. Luminescence was measured by the Enspire 2300 microplate reader. To further study the effect of compounds on bacterial growth, a suspension containing 1 × 10^8^ CFU/mL of WHQ01 or WHQ02 (0.5 McFarland standard) was diluted 1:100 with LB and added to a 96-well plate. Compounds were added to the 96-well plate at a final concentration of 100 µM and diluted in a gradient. DMSO of the same volume was added as the control group. The plate was grown at 37 °C for 18–24 h. The OD_600_ values were recorded using the Enspire 2300 microplate reader. The experiment was performed three times.

## 4. Conclusions

Of all known types of bacterial QS, only AI-2 QS exists in both Gram-positive and Gram-negative bacteria. However, research on AI-2 QSIs is still in its infancy, and the LsrK inhibitors reported at present are ATP competitive inhibitors or DPD analogues. Here, we first reported novel AI-2 QSIs targeting the LsrK/HPr PPI site. The existence of a small-molecule binding site at the LsrK/HPr PPI site and the key residues involved in the interactions between HPr and LsrK were determined by MD simulation and binding site analysis. Virtual screening was processed with the constraint of forming at least one hydrogen bond with the key amino acid residues to increase the hit rate. Positive hit compounds were discovered by the LsrK inhibition assay and the AI-2 QS interference assay. The affinity between positive hit compounds and LsrK-N protein was tested by an SPR assay to confirm that the compounds bound at the LsrK/HPr PPI site. SARs emphasized the importance of hydrophobic interactions with hydrophobic pockets, and hydrogen bonds or salt bridges with polar residues. Of the discovered AI-2 QSIs, 4171-0375, 4929-0003, and K659-0421 demonstrated similarities in both molecular docking results and their chemical structures, and they all exhibited high LsrK inhibition and AI-2 QS inhibition. Among them, 4171-0375 demonstrated the advantages of easy synthesis and high LsrK inhibition, and is suitable as a lead compound to find more effective AI-2 QSIs. It is noteworthy that HPr serves as an interacting global regulator of carbon and energy metabolism and regulates the activities of multiple proteins through direct protein–protein interactions [45,46,47,48]. We believe that it is promising to develop multi-target inhibitors through structural modification of existing LsrK/HPr PPI inhibitors.

## Figures and Tables

**Figure 1 pharmaceuticals-16-00737-f001:**
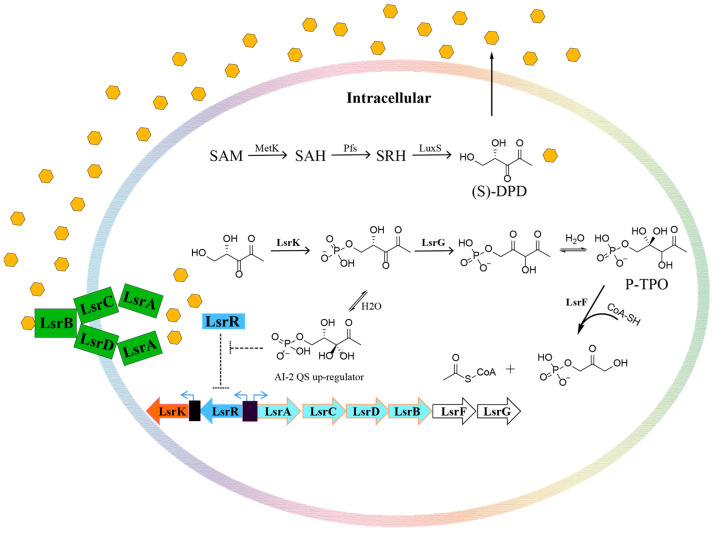
The pathway of AI-2 quorum sensing.

**Figure 2 pharmaceuticals-16-00737-f002:**
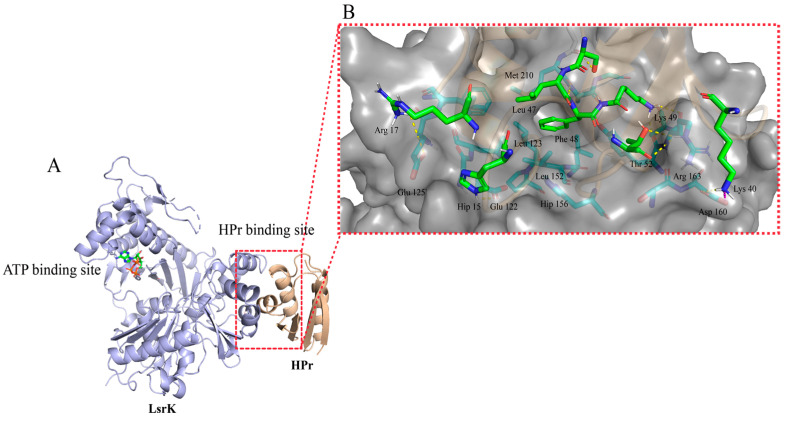
Crystal structure of the LsrK/HPr complex and close-up view of the LsrK/HPr PPI site. (**A**) Crystal structure of the LsrK/HPr complex. The LsrK and HPr are represented as purple and gold cartoons, respectively. ATP is represented as a stick. (**B**) Close-up view of the LsrK/HPr PPI site. The surface of LsrKis represented in dark gray. The key residues involved in the interactions between LsrK and HPr are represented as sticks. Yellow lines represent hydrogen bonds. Purple lines represent salt bridges.

**Figure 3 pharmaceuticals-16-00737-f003:**
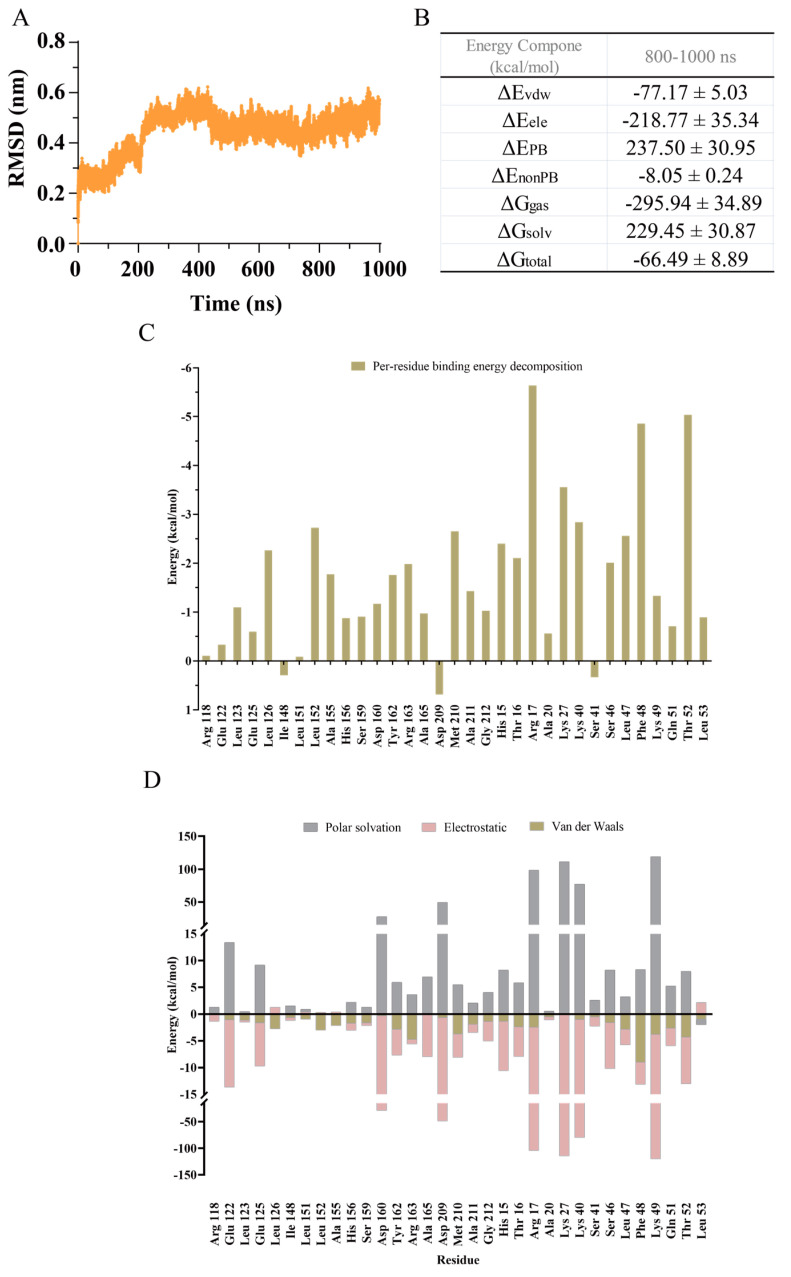
MD simulation of the LsrK/HPr complex during 1000 ns. (**A**) RMSD values of the LsrK/HPr backbone. (**B**) Calculated binding free energy of the LsrK/HPr complex during 800–1000 ns. ΔE_vdw_: contribution of the van der Waals energy; ΔE_ele_: contribution of the electrostatic energy; ΔE_PB_: contribution of the polar solvation energies; ΔE_nonPB_: contribution of the nonpolar solvation energies; ΔG_gas_: contribution of ΔE_vdw_ + ΔE_ele_; ΔG_slov_: contribution of ΔE_PB_ + ΔE_nonPB_; ΔG_total_: the final estimated binding free energy from ΔG_gas_ + ΔG_slov_. (**D**) Per-residue free energy decomposition of key residues involved in HPr binding with LsrK. (**C**) Per-residue free energy decomposition of the LsrK/HPr complex. (**D**) For each residue in the LsrK/HPr complex, the energy was also decomposed into polar solvation, electrostatic, and Van der Waals components. The binding energy was decomposed by the MM/PBSA method.

**Figure 4 pharmaceuticals-16-00737-f004:**
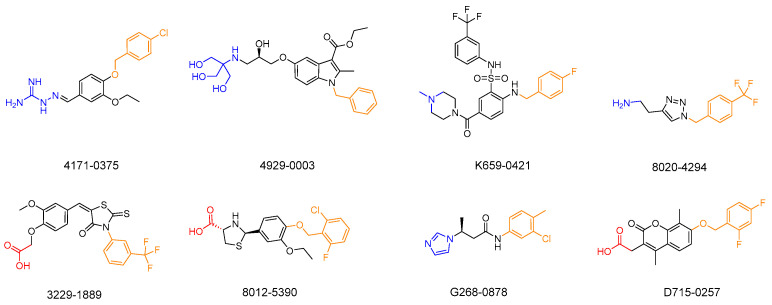
Chemical structures and compound IDs of the eight positive hits from the LsrK inhibition assay. Based on the results of ligand docking, the hydrophobic group inserted into the hydrophobic pocket is indicated in yellow, and the polar group interacting with the polar amino acid residue is indicated in blue (alkaline group) or red (acidic group).

**Figure 5 pharmaceuticals-16-00737-f005:**
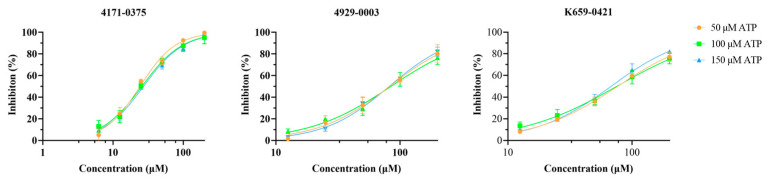
Determination of the dose-response curves for compounds 4171-0375, 4929-0003, and K659-0421 at different ATP concentrations (50, 100, and 150 μM).

**Figure 6 pharmaceuticals-16-00737-f006:**
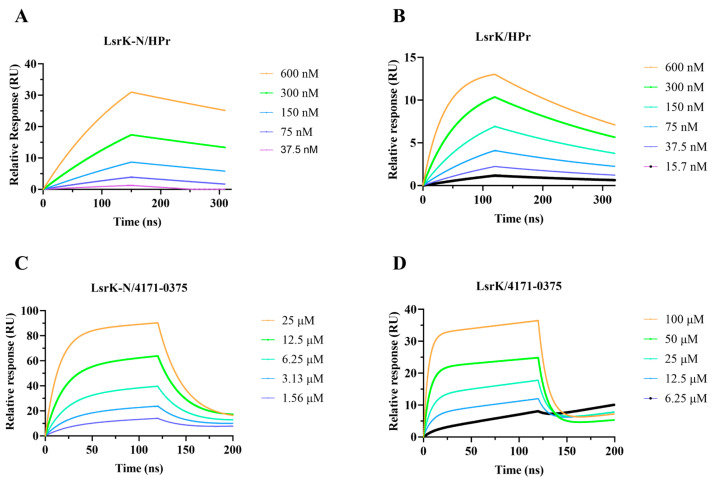
The surface plasmon resonance (SPR) results for LsrK-N/HPr (**A**), LsrK/HPr (**B**), LsrK-N/4171-0375 (**C**), and LsrK/4171-0375 (**D**). The combination and dissociation times were 120 s and 200 s, respectively, for both LsrK-N/HPr and LsrK/HPr. The results displayed that HPr slowly bound to and slowly dissociated from LsrK or LsrK-N. The combination and dissociation times were 120 s and 80 s, respectively, for both LsrK-N/4171-0375 and LsrK/4171-0375. The results indicated that 4171-0375 quickly bound and dissociated from LsrK or LsrK-N.

**Figure 7 pharmaceuticals-16-00737-f007:**
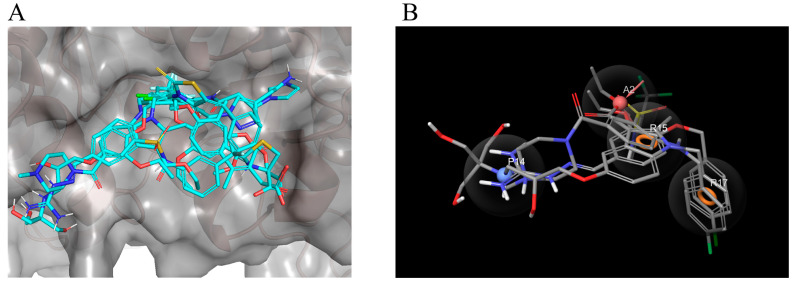
Docking results of eight positive hits and the analysis of pharmacophore. (**A**) Superposition of the ligand docking results for the eight positive hits in the LsrK inhibition assay. The surface of LsrK is indicated in dark gray. The key residues involved in the interactions between LsrK and the hit compounds are indicated as sticks. (**B**) A depiction of the top-score pharmacophore model with its reference ligands.

**Figure 8 pharmaceuticals-16-00737-f008:**
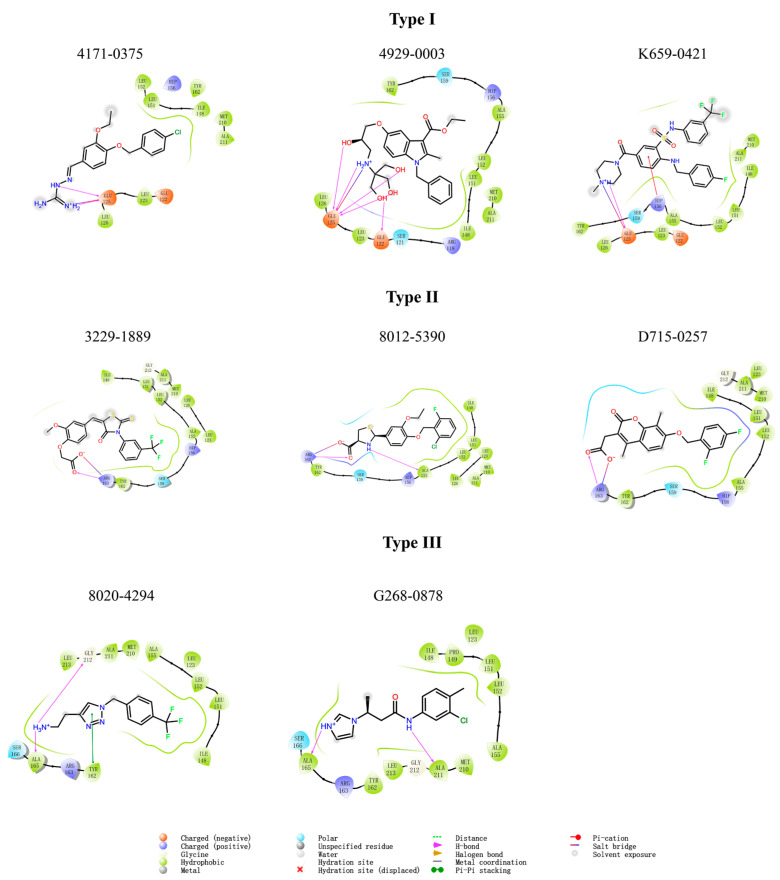
Two-dimensional ligand docking results for the eight positive hits.

**Table 1 pharmaceuticals-16-00737-t001:** IC_50_ values of the eight positive hits from the LsrK inhibition assay and the cell-based AI-2 QS interference assay.

Compound ID	LsrK Inhibition (%)	LsrK (IC_50_) ^A^	WHQ02 (IC_50_) ^B^	WHQ01 (IC_50_) ^C^
4171-0375	98 ± 7	26.13 ± 2.94	30.13 ± 1.62 *	42.93 ± 1.63 *
4929-0003	78 ± 12	80.75 ± 4.30	35.40 ± 2.29 *	20.89 ± 1.24 *
K659-0421	75 ± 8	73.69 ± 5.33	16.98 ± 0.37 *	7.97 ± 0.38 *
8020-4294	65 ± 9	148.40 ± 6.92	67.25 ± 4.32	24.86 ± 1.10
3229-1889	63 ± 4	121.31 ± 5.37	14.99 ± 1.24	24.72 ± 1.42
8012-5390	60 ± 4	147.13 ± 5.58	17.68 ± 0.98	23.28 ± 1.05
G268-0878	58 ± 5	128.73 ± 6.14	42.30 ± 1.42	30.55 ± 1.26
D715-0257	57 ± 7	152.57 ± 5.43	22.34 ± 2.15	21.10 ± 1.18

^A^—IC_50_ values of the compounds from the LsrK inhibition assay. ^B^—IC_50_ values of the compounds from the cell-based AI-2 QS interference assay in WHQ02 (*E. coli* BL21 ΔTolC pWHQ01). ^C^—IC_50_ values of the compounds from the cell-based AI-2 QS interference assay in WHQ01 (*S. typhimurium* ATCC 202165 ΔTolC pWHQ01). The units for all IC_50_ values were micromoles. IC_50_ values and LsrK inhibition rates represented the means ± SD from three independent experiments (*n* = 3). *—indicates that a compound demonstrated obvious inhibition of bacterial growth in this assay.

## Data Availability

Data is contained within the article and the Appendix A.

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
