# Peer review of "Discovery of AI-2 Quorum Sensing Inhibitors Targeting the LsrK/HPr Protein–Protein Interaction Site by Molecular Dynamics Simulation, Virtual Screening, and Bioassay Evaluation"

_pharmaceuticals, 2023, doi:10.3390/ph16050737_

Round 1
Reviewer 1 Report
Summary of the key contribution of the paper:
The Discovery of AI-2 quorum sensing inhibitors targeting the LsrK/HPr protein-protein interaction site by molecular dynamics Simula describes the structure-activity relationships (SARs) emphasized the importance of hydrophobic interactions with the hydrophobic pocket and hydrogen bonds or salt bridges with key residues of LsrK for LsrK/HPr PPI inhibitors. These new AI-2 QSIs, especially 4171-0375, exhibited novel structures, significant LsrK inhibition and were suitable for structural modification to search for more effective AI-2 QSIs, the scope of the manuscript is very interesting, it is promising to develop multi-target inhibitors through structural modification of existing LsrK/HPr PPI inhibitors, the manuscript can be accepted after addressing the following concerns.
Highlights:
1) The manuscript provides the advantages of easy synthesis and high LsrK inhibition and is suitable as a lead compound to find more effective AI-2 QSIs. It is noteworthy that HPr serves as an interacting global regulator of carbon and energy metabolism and regulates the activities of multiple proteins through direct protein–protein interactions.
2) The experimental results show research on AI-2 QSIs is still in its infancy, and the LsrK inhibitors reported at present are ATP competitive inhibitors or DPD analogues, so they first reported novel AI-2 QSIs targeting the LsrK/HPr PPI site. The existence of a small-molecule binding site at the LsrK/HPr PPI site and the key residues involved in the interactions between HPr and LsrK were determined by MD simulation and binding site analysis.
3) The quorum sensing (QS), AI-2 QS exists in both Gram-negative and Gram-positive bacteria and is responsible for interspecies communication.
4) Quorum sensing is a cell-to-cell communication mechanism that regulates bacterial pathogenicity, biofilm formation, and antibiotic sensitivity.
5) The figures and tables are well referenced and clear.
6) The resulting manuscript will greatly contribute to academia as well as industry.
Lowlights: There are no Lowlights in this paper
Author Response
Dear Reviewer,
Thank you for reviewing our manuscript and for your valuable time and effort in providing feedback. We appreciate your positive feedback and are pleased to hear that our research meets the standards of the journal.
We would like to thank you and the editorial team for the opportunity to publish our work in Pharmaceuticals. We are grateful for your help and support in improving the quality of our work.
Thank you again for your review and we look forward to hearing from you.
Best regards,
Yijie Xu
Reviewer 2 Report
The authors propose novel quorum-sensing inhibitors designed for bacterial systems. The techniques used for designing the novel molecules were computational and experimental. The manuscript is well written. However, there are some issues:
-the introduction should be expanded with details regarding quorum sensing theory(the ability to detect and respond to cell population density by gene regulation.), and its application to bacterial systems. ( This process is highly dependent on the diffusion mechanism of the signaling molecules) ..
Line 103, the binding site coordinates should be specified
-the database base used for screening should be described in a few lines ( add a list with the compounds used for screening -if not 1.6 mil compounds, at least some of them – so the others can reproduce the study)
-why the 62 molecules were chosen – please explain the criteria more clearly
-docking studies should be validated statistically
-if Schrodinger was used, what is the pharmacophore hypothesis -please discuss it ( figure 7 B)
-if experimental data and computational data were available, why a QSAR model was not generated?
Author Response
Dear Reviewer,
Thank you for reviewing our manuscript and for your valuable time and effort in providing feedback. We would like to thank you and the editorial team for the opportunity to publish our work in Pharmaceuticals. We are grateful for your help and support in improving the quality of our work. We have carefully considered your comments and made appropriate changes to the manuscript according to your suggestions. Please find attached the revised manuscript with a point-by-point response to each of your comments.
Thank you again for your review and we look forward to hearing from you.
Best regards,
Yijie Xu

Reviewer 3 Report
In this manuscript, Yijie Xu et al described some novel AI-2 quorum sensing inhibitors targeting the LsrK/HPr interaction site. The authors performed different techniques to find the LsrK/HPr binding pocket, such as Virtual Screening Workflow (VSW), Docking, and SPR assay, among others. All of these results showed a new binding site, which could be targeted by small molecules.
Then, the authors carried out a screening to search for potential LsrK/HPr inhibitors from the commercial compound library ChemDiv (1.6 million molecules) to eventually obtained 62 commercially available compounds.
All of these compounds were tested as LsrK inhibitors and in a Cell-based AI-2-mediated QS interference assay. Finally, 8 of those compounds were selected as the most potent inhibitors. Of note, one of these compounds inhibited Lsrk in a 99% and at low micromolar IC.
The authors describe for first time specific Lsrk inhibitors, which neither are ATP competitive nor DPD derivative.
However, the work has some weak points. All of these molecules are not new. They have been described for other applications. Furthermore, some works have described novel QS inhibitors, although following a different mechanism of action but they have more potency than all described in this work.
Anyway, the reviewer considers this a great work to be published in this journal, specially becuase the authors have described through different and compatible techniques the binding pocket of LsrK/HPr, opening the door to explore new molecules that target this binding site to improve their activity.
Author Response

(The authors gave the same response as above.)

Round 2
Reviewer 2 Report
The manuscript can be published in its present form